# Utilization of cervical cancer screening and associated factors among women in Debremarkos town, Amhara region, Northwest Ethiopia: Community based cross-sectional study

**Bewket Yeserah Aynalem**[1]*, **Kiber Temesgen Anteneh**[2], **Mihretu Molla Enyew**[2]

**1** Department of Midwifery, Debremarkos University, Debremarkos, Ethiopia, **2** Department of Midwifery, University of Gondar, Gondar, Ethiopia

* by123bewket@gmail.com

## Abstract

### Introduction

Cervical cancer is the most common type of malignancy among all malignancies for women worldwide with 266 000 deaths every year. Even though there is a proven importance of cervical cancer screening, the death of women due to cervical cancer in Ethiopia is high. We, therefore, did this study to investigate the utilization of cancer screening and its associated factors among women in Debremarkos town, Amhara region, Ethiopia.

### Methods

A community-based cross-sectional study was conducted among women from 30–49 years in Debremarkos town, from July 1 to August 30, 2018. A multistage sampling procedure was used to select 822 women in the study. We used EPI info version 7 for data entry and SPSS version 24 software for cleaning and analysis. Bivariable and multivariable logistic regression analyses were performed to identify factors associated with the utilization of cervical cancer screening. Variables with a p-value of less than 0.05 were taken as significant variables.

### Result

The study revealed that 44 (5.4%) of women have been screened for cervical cancer. Women's age [AOR:3.126(1.246,7.845)], marital status (AOR:3.41(1.299,8.972)], educational status(secondary education level [AOR: 4.578(95% CI: 1.19, 17.65)] and College and above education level [AOR:7.27,95%CI: 2.07,25.513)]), started sexual intercourse for the first time below 16 years[AOR:3.021(1.84,4.97)], history of multiple sexual partners [AOR:2.51(1.040, 6.06)], history of sexually transmitted disease [AOR:4.04(1.68, 9.72),], knowledge on cervical cancer screening [AOR:4.02(2.07,7.77)] and attitude towards

**Data Availability Statement:** All relevant data are within the paper and its Supporting Information files.

**Funding:** The author(s) received no specific funding for this work.

**Competing interests:** The authors have declared that no competing interests exist.

cervical cancer screening [AOR:3.23(2.52,4.12)] were significant factors for utilization of cervical cancer screening

## Conclusion

This study showed the magnitude of the utilization of cervical cancer screening is very low. Women's age, marital status, educational status, age at first sex history of multiple sexual partners and sexually transmitted disease, knowledge and attitude were important factors of screening. Therefore, intervention programs that are aimed at improving cervical cancer screening practice among women should focus on the identified factors.

## Introduction

Cervical cancer is the most common malignancy among all malignancies for women worldwide [1]. In 2012, about 266, 000 deaths occurred due to cervical cancer worldwide and 90% of cases occurred in Low and Middle-Income Countries (LMICs); the highest was occurred in Sub Saharan Africa (SSA) where cervical cancer is the leading killer among women[2]. In Ethiopia,7095 cases and 4732 death of cervical cancer occur every year [3].

Cervical cancer screening has been proven to be a very effective prevention strategy for cervical cancer [1, 4, 5]. The United States Preventive Services Task Force (USPSTF), the American Cancer Society (ACS) World Health Organization (WHO) guidelines state screening for cervical cancer once within lifetime reduces significantly the risk of mortality from cervical cancer and incidence of advanced cervical cancer [6–8]. The WHO recommended age for cervical cancer screening should be limited to women the age of 30 to 50 years since there is evidence on younger age women with a milder degree of lesion spontaneously recover to normal [5, 9, 10].

Ethiopia adopted the WHO guideline and advised women to start cervical cancer screening at age of 30–49 years at least one to three years interval with the approach of seeing and treat through Visual Inspection under Acetic acid (VIA) as screening strategy and cryo-therapy as a treatment method[1, 4, 5, 11].

Despite the importance of screening, a high incidence of cervical cancer is still a big problem and a major cause of morbidity and mortality of women in LMICs especially in SSA [12, 13] and the lowest cervical cancer screening rate found in Ethiopia which accounts below 1% [14].

Therefore, the main aim of this research was to identify factors affecting cervical screening utilization and recommend ways to increase screening utilization by the community.

## Methods

### Study area and period

This study was conducted in Debremarkos town, Amhara region, Northwest Ethiopia from July 1 to August 30, 2018. Debremarkos town is the capital city of East Gojjam Zone that is found in the Amhara region, North West Ethiopia, which is located at 300 km from Addis Ababa, the capital city of Ethiopia and 265 km from Bihar Dar, the capital city of Amhara region. There are one referral hospital, three health centers and five non-governmental clinics that give different reproductive health services in the town. Only the government referral hospital gives cervical cancer screening services.

## Study design

A community-based cross-sectional study design was employed.

## Study participants

The source population was all women age 30–49 years old who were residents of Debremarkos town. The study population was women age 30–49 years old who were residents of Debremarkos town during the study period in the selected kebeles. We excluded women who are not permanent residents of the town (less than six months) and those who were critically ill during the data collection period.

## Sample size

The sample size for prevalence was determined based on a single population proportion formula assumption. The expected proportion of cervical cancer utilization (19.8%) from the previous study in Ethiopia at Mekelle town[15]and a 3.5%confidence limit (margin of error) was used.

$$initial sample size = \left(Z\frac{a}{2}\right)^2 * \frac{p(1-p)}{w2} = 1.96^2 * \frac{0.198(1-0.198)}{(0.035)^2} = 498$$

With considering design effect 1.5 since it had two stages and the sample size was calculated as498*1.5 = 747then the non-response rate was also considered to be 10%and 747*0.10 = 75. Then the final sample size was747+75 = 822.

## Sampling techniques

A multistage sampling technique was used and firstly all the kebeles found in the Debremarkos town were listed in a frame. Then three out of the seven kebeles were selected by the lottery method. Again the list of households found and coded in each kebele. The size of households consisting of eligible population to be selected from each kebele was determined proportionally based on the size of the study units and the $k^{th}$ value was computed for each selected kebele. The woman of the selected household was interviewed and if there was more than one woman in the household, the lottery method has been used to select only one. In the case of absenteeism, after three repeated visits the next eligible woman was included in the study.

## Study variables

**Dependent variable.** Utilization of cervical cancer screening

**Independent variables.** Socio-Demographic Characteristics, Reproductive and behavioral characteristics, and knowledge and attitude on cervical cancer and its screening.

## Operational definitions

**Utilization of cervical cancer screening.** Refers to the proportion of persons eligible to be screened within a population who have been screened within 3 years for cervical cancer[15].

**Multiple sexual partners.** Those women who have ever had penetrative sexual intercourse with more than one partner in their life serially or at the same time [16].

**Cigarette smoking.** The active smoking or ever had a smoking history of women one or more manufactured or hand-rolled tobacco cigarettes per day which excludes passive smokers [17].

**Knowledgeable.**   Women who answered knowledge questions score of mean value or above were considered as knowledgeable.

**Favorable attitude.**   Women who answered attitude questions a score of mean value or above were considered to have a favorable attitude.

### Data collection and data quality control

To assure the data quality, data were collected with face to face interviews by three trained BSc Midwives after one-day data collection training was given to them together with three MSc holder supervisors. The questionnaire was structured and pre-tested which was first prepared in English and translated to local (Amharic) language and then again translated back to English. A pretest was conducted on 42 women of the sample size in other than the study area and the necessary correction on the tool was employed accordingly.

### Data processing and analysis

Epi Info version 7 software was used for data entry and SPSS version 24 for used for analysis. Bivariate logistic regression was employed to identify an association between independent and dependent variables. Variables having a P-value of less than 0.2 in the bivariate logistic regression analysis were fitted into the multivariable logistic regression model. The 95% confidence interval of odds ratio was computed and variable having P-value less than 0.05 in the multivariable logistic regression analysis was considered as statistically significant.

### Ethical clearance

Ethical clearance was obtained from the Ethical Review Committee of the Department of Midwifery, under the delegation of the institutional review board of the University of Gondar. Ethical clearance and formal letters were also obtained from the University Gondar School of Midwifery and were submitted to Debremarkos health office and permission was obtained. Finally, written informed consent was also obtained from each study participant.

## Results

### Socio-demographic characteristics of the respondents

All 822 study participants responded to the questionnaire, giving a response rate of 100%. The mean age of the study participants was 36.81 years (36.81 ±5.14 SD). The majorities of women were Amhara 809 (98.4%), Christian religion 789 (96%) and married 768(64.8%). Two hundred thirty-one (28.1%) of the women have not attended formal education (Table 1).

### Reproductive and behavioral characteristics

Two hundred one (24.5%) study participants first had sexual intercourse at age 16 and below. Women who had a history of multiple sexual partners (MSPs) within three years were 461 (56.1%). And 129 (15.7%) had a history of sexually transmitted disease (STD). Six hundred eighty-nine (83.8%) of the study participants had used modern family planning method at least for one year. Six hundred sixty-three (80.7%) got birth at least once. Around ninety-seven (11.8%) of respondents had a family history of cervical cancer and 17 (2.1%) of the respondents had also smoking history (Table 2).

**Table 1. Sociodemographic characteristics of women (n = 822) age 30–49 in Debremarkos town, Northwest Ethiopia, 2018.**

| Variable | Frequency | Percent |
|---|---|---|
| **Age of women** | | |
| 30–39 | 592 | 72 |
| 40–49 | 230 | 28 |
| **Marital status** | | |
| Married | 638 | 77.6 |
| Others* | 184 | 22.4 |
| **Religion** | | |
| Christian | 789 | 96 |
| Muslim | 33 | 4 |
| **Educational status** | | |
| No formal education | 231 | 28.1 |
| Primary education | 180 | 21.9 |
| Secondary education | 212 | 25.8 |
| College and above | 199 | 24.2 |
| **Ethnicity** | | |
| Amhara | 809 | 98.4 |
| Others** | 13 | 1.6 |
| **Occupation** | | |
| Housewife | 149 | 18.1 |
| Self-employee (doing own small business) | 309 | 37.6 |
| Private employee(salaried in the nongovernmental sector) | 192 | 23.4 |
| Government employee | 172 | 20.9 |
| **Household income** *** | | |
| <900 | 166 | 20.2 |
| 900–1600 | 201 | 24.5 |
| 1601–2699 | 293 | 35.6 |
| > = 2700 | 162 | 19.7 |

*Single, divorced and widowed,

**Oromo and Gurage,

***in Ethiopian Birr

## Knowledge, attitude and utilization of cervical cancer screening

More than half of the respondents (59%) were knowledgeable about cervical cancer screening (Table 3). More than half of the respondents (57.4%) had a favorable attitude towards cervical cancer screening (Table 4). Forty-four (5.4%) of the study population have been utilized cervical cancer screenings a minimum of once within the last three years with [95% CI: 3.8, 7.1] (Table 5).

## Factors associated with utilization of cervical cancer screening

After controlling the effect of other variables with binary logistic regression, age, marital status, educational status, age start at first sexual intercourse, history of MSP, history of STD, duration of FP use, family history of CC, gravidity, knowledge, and attitude about CC screening continued to be significantly associated with utilization of cervical cancer screening (P-values<0.2).

After controlling the effect of other variables with multivariable logistic regression analysis, women's age [AOR: 3.126 (95% CI: 1.246, 7.845)], marital status [AOR: 3.41 (95% CI: 1.299,

**Table 2. Reproductive characteristics of women (n = 822) age 30–49 in Debremarkos town, Northwest Ethiopia, 2018.**

| Variable | Frequency | Percent |
|---|---|---|
| **Age started sexual intercourse (in years)** | 201 | 24.5 |
| < = 16 | | |
| >16 | 621 | 75.5 |
| **Multiple sexual partner** | | |
| No | 461 | 56.1 |
| Yes | 361 | 43.9 |
| **History of smoking** | | |
| No | 805 | 97.9 |
| Yes | 17 | 2.1 |
| **History of STD** | | |
| No | 693 | 84.3 |
| Yes | 129 | 15.7 |
| **Ever use a modern FP method** | | |
| No | 133 | 16.2 |
| Yes | 689 | 83.8 |
| **Duration of modern FP method use** | | |
| 1–4 years | 530 | 76.9 |
| > = 5 years | 159 | 23.1 |
| **Family history of cervical cancer** | | |
| No | 725 | 88.2 |
| Yes | 97 | 11.8 |
| **Ever had got pregnant** | | |
| No | 127 | 15.5 |
| Yes | 695 | 84.5 |
| **Gravidity** | | |
| 1–5 | 529 | 76.1 |
| >5 | 166 | 23.9 |
| **Ever had given birth** | | |
| No | 159 | 19.3 |
| Yes | 663 | 80.7 |
| **Parity** | | |
| 1–5 | 558 | 84.2 |
| >5 | 105 | 15.8 |

8.972)], education level (secondary education level [AOR: 4.578(95% CI: 1.19, 17.65)] and College and above education level [AOR:7.27,95%CI: 2.07,25.513)]), age started sexual intercourse for the first time [AOR:3.021(95%CI:1.84, 4.97)], history of MSP[AOR:2.51 (95% CI:1.040, 6.06)], history of sexually transmitted disease [AOR:4.04(95% CI:1.68, 9.72)], knowledge about cervical cancer screening [AOR:4.02,95% CI:2.07, 7.77)] and attitude about cervical cancer screening [AOR:3.23,95%CI;2.52, 4.12)] were significant factors for utilization of cervical cancer screening (Table 6).

## Discussion

This study was conducted to assess the cervical screening practice among women in Debremarkos town, northwest Ethiopia. Accordingly, the study found that only 5.4% of women had cervical cancer screening practice. Similarly, factors like women's age, marital status, education

**Table 3. Knowledge about cervical cancer screening among women (n = 822) in Debremarkos town, Northwest Ethiopia, 2018.**

| Variable | | Frequency | Percent |
|---|---|---|---|
| **Ever heard about cervical cancer** | | | |
| No | | 130 | 15.8 |
| Yes | | 692 | 84.2 |
| **Ever heard about cervical cancer screening** | | | |
| No | | 130 | 15.8 |
| Yes | | 692 | 84.2 |
| **All eligible women can have cervical cancer screening without complication** | | | |
| No | | 327 | 39.8 |
| Yes | | 495 | 60.2 |
| **Knew health institutions that give cervical cancer screening service** | | | |
| No | | 388 | 47.2 |
| Yes | | 434 | 52.8 |
| **Knew symptoms of cervical cancer** | | | |
| No | | 740 | 90.0 |
| Yes | | 82 | 10.0 |
| **Bleeding during sexual intercourse may be one of the signs of cervical cancer** | | | |
| No | | 740 | 90.0 |
| Yes | | 82 | 10.0 |
| **Cervical cancer is a killer disease** | | | |
| No | | 281 | 34.2 |
| Yes | | 541 | 65.8 |
| **Is cervical cancer preventable disease** | | | |
| No | | 379 | 46.1 |
| Yes | | 443 | 53.9 |
| **Is cervical cancer curable disease** | | | |
| No | | 366 | 44.5 |
| Yes | | 456 | 55.5 |
| **May have cervical cancer without any sign and symptom** | | | s |
| No | | 480 | 58.4 |
| Yes | | 342 | 41.6 |
| **Overall knowledge** | | | |
| | Knowledgeable | 485 | 59 |
| | Not Knowledgeable | 337 | 41 |

level, and age at first sexual intercourse below 16 years old, history of STDs, knowledge, and attitude towards cervical cancer screening were significantly associated with utilization of cervical cancer screening.

The study finding of cervical cancer screening utilization (5.4%) in the current study was in-line with similar reports in Arbaminch town, Southern Ethiopia(5.9%)[18]. The finding of this research was lower than the studies done at Addis Ababa, Ethiopia(10.8%) [19], at Mekelle town, Northern Ethiopia(19.8%) [15], at Hadiya Zone, South Ethiopia (9.9%)[20], in Ethiopia (17%) [21], Kenya(16%)[22]and Nigeria (11%) [23].The possible explanation for this might be the difference in the age of study population; the difference in the study area; the difference in

**Table 4. Attitude towards cervical cancer screening among women (n = 822) in Debremarkos town, Northwest Ethiopia, 2018.**

| Variables | Level of agreement | | | | | |
|---|---|---|---|---|---|---|
| | Agree | | Disagree | | Indifferent | |
| | Number | Percent | Number | Percent | Number | Percent |
| Any reproductive age woman is susceptible to develop cervical cancer | 390 | 47.4 | 225 | 27.4 | 207 | 25.2 |
| Like any women, you are susceptible to develop cervical cancer | 372 | 45.3 | 248 | 30.2 | 202 | 24.6 |
| Cervical Cancer can be transmitted genetically | 111 | 13.5 | 454 | 55.2 | 257 | 31.3 |
| Cervical cancer may be dangerous | 481 | 58.9 | 180 | 22 | 156 | 19.1 |
| Precancerous cervical screening may be beneficial to health | 481 | 58.9 | 180 | 22 | 156 | 19.1 |
| Cervical cancer screening is painful | 436 | 53.4 | 153 | 18.6 | 233 | 28.3 |
| **Overall attitude** | **Frequency** | | | | **Percent** | |
| Favorable attitude | 474 | | | | 57.4 | |
| Unfavorable attitude | 348 | | | | 42.6 | |

the health status of the study population; the difference with sample size; the difference with the level of knowledge and attitude of the study population.

The finding of this study was also higher than studies done in Ethiopia (2.9%) [17] and Southern Ghana (0.8%) [24]. The possible explanation might be the difference in the study area and study population; the difference in the age of the study population.

As shown in this study, women's age was one of the significant factors for the utilization of cervical cancer screening. Women in their age 40–49 years were 3.126 times more likely to utilize cervical cancer screening as compared to women in their age of 30–39 years [AOR:3.126 (95% CI:1.246, 7.845)]. This finding was supported by the studies done in Northern Ethiopia, Addis Ababa, Southern Ethiopia, Ethiopia and Malawi [15, 17, 18, 25, 26]. The possible explanation for this might be like women's age increases the probability of getting information about cervical cancer and its screening will be increased which leads them utilized cervical cancer screening service. The other explanation also might be increasing risk with women's age leads the women to have more contact health facilities.

Marital status was also one of the significant predictors for the utilization of cervical cancer screening. This study showed that women who were single/divorced/widowed 3.414 times more likely utilized cervical cancer screening as compared with married women [AOR:3.414, (95%CI:1.299,8.972)]. This result was supported by the study done in Thailand [27]. The possible explanation for this might be single women are more likely educated since they might be young and divorced women or widowed women are a more likely aged and increasing risk with women's age leads the women to have more interest to visit health facilities.

Educational status was also the main significant factor for the utilization of cervical cancer screening. Women who took college & above education 7.268 times more likely utilized as

**Table 5. Utilization of cervical cancer screening among women (n = 822) in Debremarkos town, Northwest Ethiopia, 2018.**

| Screened at least once during the last three years | | Frequency | Percent |
|---|---|---|---|
| | Yes | 44 | 5.4 |
| | No | 778 | 94.6 |

**Table 6. Bivariable and multivariable analysis of factors associated with utilization of cervical cancer screening among women in Debremarkos town, Northwest Ethiopia, 2018.**

| Variable | Utilized | | Crude OR[95%CI] | AOR[95%CI] |
|---|---|---|---|---|
| | **Yes** | **No** | | |
| **Age of women** | | | | |
| 40–49 | 31 | 199 | **6.938(3.56,13.52)** | **3.126(1.246,7.8)**[*] |
| 30–39 | 13 | 579 | **1** | **1** |
| **Marital status** | | | | |
| Others** | 15 | 169 | **1.864(1.09,3.557)** | **3.41(1.299,8.97)**[*] |
| Married | 29 | 609 | **1** | **1** |
| **Religion** | | | | |
| Muslim | 1 | 32 | 0.542(0.072,4.062) | |
| Christian | 43 | 746 | 1 | |
| **Educational status** | | | | |
| Primary education | 7 | 173 | 1.295 (.446,3.761) | |
| Secondary education | 10 | 202 | **1.584(1.059,4.240)** | **4.578(1.19,17.6)**[*] |
| College and above | 20 | 179 | **3.575(1.479,8.645)** | **7.27(2.07,25.51)**[*] |
| No formal education | 7 | 224 | **1** | **1** |
| **Ethnicity** | | | | |
| Others*** | 1 | 12 | 1.484(.189,11.682) | |
| Amhara | 43 | 766 | 1 | |
| **Occupation** | | | | |
| Self-employee | 14 | 295 | 1.131(1.426,3.005) | |
| Private employee | 16 | 176 | 2.167(.826,5.681) | |
| Government employee | 8 | 164 | 1.163(.394,3.430) | |
| Housewife | 6 | 143 | 1 | |
| **Household income**[****] | | | | |
| 900–1600 | 10 | 191 | 1.686(.565,5.033) | |
| 1601–2699 | 18 | 275 | 2.108(.768,5.785) | |
| > = 2700 | 11 | 151 | 2.346(.796,6.909) | |
| <900 | 5 | 161 | 1 | |
| **Age started sexual intercourse for the first time** | | | | |
| < = 16 | 24 | 177 | **4.075(2.483,6.717)** | **3.021(1.84,4.97)**[*] |
| >16 | 20 | 601 | **1** | **1** |
| **Multiple sexual partner** | | | | |
| Yes | 30 | 331 | **2.894(1.511,5.544)** | **2.511(1.04,6.06)**[*] |
| No | 14 | 447 | **1** | **1** |
| **History of smoking** | | | | |
| Yes | 2 | 15 | 2.422(.536,10.94) | |
| No | 42 | 763 | 1 | |
| **History of STD** | | | | |
| Yes | 18 | 111 | **4.160(2.208,7.840)** | **4.037(1.68,9.72)**[*] |
| No | 26 | 667 | **1** | **1** |
| **Ever use a modern FP method** | | | | |
| Yes | 34 | 655 | .638(.307,1.326) | |
| No | 10 | 123 | 1 | |
| **Duration of modern FP method usage** | | | | |
| > = 5 years | 18 | 141 | 3.433(1.746,6.68) | 1.771(.711,4.41) |
| 1–4 years | 19 | 511 | 1 | 1 |

*(Continued)*

**Table 6.** (Continued)

| Variable | Utilized | | Crude OR[95%CI] | AOR[95%CI] |
|---|---|---|---|---|
| | Yes | No | | |
| **Family history of cervical cancer** | | | | |
| Yes | 8 | 89 | 1.720(1.041,4.218) | 1.897(.701,5.134) |
| No | 36 | 689 | 1 | 1 |
| **Ever had got pregnant** | | | | |
| Yes | 39 | 656 | 1.451 (.517,3.101) | |
| No | 5 | 122 | 1 | |
| **Gravidity** | | | | |
| >5 | 19 | 147 | 3.469(1.790, 6.72) | 1.421(.58, 3.47) |
| 1–5 | 19 | 510 | 1 | 1 |
| **Ever had given birth** | | | | |
| Yes | 35 | 628 | .929(.437, 1.974) | |
| No | 9 | 150 | 1 | |
| **Parity** | | | | |
| >5 | 7 | 98 | 1.257(.540,2.958) | |
| 1–5 | 30 | 528 | 1 | |
| **Knowledge** | | | | |
| Knowledgeable | 31 | 454 | **1.702(1.01,3.29)** | **4.02(2.07, 7.77)*** |
| not knowledgeable | 13 | 324 | **1** | **1** |
| **Attitude** | | | | |
| favorable attitude | 33 | 439 | **2.317(1.81, 2.96)** | **3.225(2.52,4.12)*** |
| unfavorable attitude | 11 | 339 | **1** | **1** |

*p-value less than 0.05,

**single, divorced and widowed,

***Oromo and Gurage,

****in Ethiopian Birr

compared with women who did not take formal education [AOR: 7.268(95% CI:2.071,25.513)].Similarly, Women who attended secondary education 4.578 times more likely utilized as compared with women who did not attend formal education [AOR: 4.578 (95% CI: 1.187,17.649)]. This study was supported by the study done in Ethiopia and Kenya [17, 22]. This might be explained with, as the level of education increases, the women will have the chance to know about cervical cancer and its screening.

Age at first sexual intercourse was a significant predictor for the utilization of cervical cancer screening. Women who had started sexual intercourse with their age of 16 years and below were 3.021 times more likely to utilize cervical cancer screening as compared to those women who had started sexual intercourse after their age of 16 years[AOR: 3.021 (95%CI; 1.84, 4.97)]. The possible explanation for this might be women who started sexual intercourse at an early age, may have increased lifetime sexual partners which in turn increase the chance of being infected with sexually transmitted infection (STI) and STD with its signs and symptoms which lead to visit health facilities. No literature found to compare with this study because it is sensitive to issue information biases might occur which was tried to avoid these biases in our study with care full interviewing technique.

This study finding showed that woman who had a history of MSPs was another important factor for utilization of cervical cancer screening. Women who have had a history of MSPs were 2.51 times more likely to utilize cervical cancer screening as compared to those who did

not MSP [AOR: 2.51(95% CI: 1.040, 6.062)]. This study was supported by the result of a study done in Addis Ababa, at Mekelle, Malawi, and Thailand [15, 25–27]. The possible explanation might be women, who had MSP history, would have the chance to be infected with STIs with its signs and symptoms which increases health facility visits.

The current study result showed that a woman who had a history of STD was another important factor in the utilization of cervical cancer screening. Women who have had a history of STD were 4.037 times more likely to utilize cervical cancer screening as compared to those who did not have STD history [AOR: 4.037, (95%Cl: 1.68, 9.72)]. This result was supported by the finding from Addis Ababa and Northern Ethiopia [15, 25]. The above association might be explained by women who have STDs and history of STD, will have an increased chance of visiting health institutions for treatment and other medical help.

Women's knowledge about cervical cancer screening was another significant factor in the utilization of cervical cancer screening services. Women who were knowledgeable about cervical cancer screening were 4.02 times more likely to utilize cervical cancer screening service as compared to those who were not knowledgeable [AOR:4.02 (95% CI:2.07, 7.77)]. This finding was supported by the result of studies done in Northern Ethiopia, Malawi, Tanzania, and Thailand [15, 26–28]. The above reports might be explained by increasing the level of women's knowledge about the benefits of screening directly lead the women to utilize cervical cancer screening.

Similarly, women's attitude about cervical cancer screening was also a significant factor for utilization of cervical cancer screening service. Women who had a favorable attitude about cervical cancer screening were 3.225 times more likely to utilize cervical cancer screening service as compared to those who had unfavorable attitudes [AOR:3.225 (95%CI:2.52, 4.12)]. This study was supported by a study done in Northern Ethiopia, Southern Ethiopia, Southern Ghana and Thailand [15, 18, 24, 27]. The above reports might be explained with women who have a favorable attitude towards cervical cancer screening will have self-initiative to know about cervical cancer risk factors and benefits of its screening.

## Conclusion

This study showed the magnitude of the utilization of cervical cancer screening was lowin Debremarkos town, Northwest Ethiopia. Age of the women, marital status, and educational status, age at first sexual intercourse, history of multiple sexual partners and sexually transmitted disease, knowledge and attitude were statistically significant factors of the utilization of cervical cancer screening.

## Acknowledgments

We would like to thank the University of Gondar and Debremarkos health office for their permission to this research and we gratefully acknowledge all study individuals for their participation in the study.

## Author Contributions

**Conceptualization:** Bewket Yeserah Aynalem.

**Data curation:** Bewket Yeserah Aynalem.

**Formal analysis:** Bewket Yeserah Aynalem, Mihretu Molla Enyew.

**Methodology:** Bewket Yeserah Aynalem, Kiber Temesgen Anteneh.

**Supervision:** Kiber Temesgen Anteneh, Mihretu Molla Enyew.

**Writing – original draft:** Bewket Yeserah Aynalem.

**Writing – review & editing:** Kiber Temesgen Anteneh, Mihretu Molla Enyew.

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
