## [Decision Letter · Decision Letter 0]

23 Sep 2019

PONE-D-19-16524

Utilization of Cervical Cancer Screening and Associated Factors among Women in Debremarkos town, Amhara Region, Northwest Ethiopia: Community based cross sectional study

PLOS ONE

Dear mr yeserah,

Thank you for submitting your manuscript to PLOS ONE. After careful consideration, we feel that it has merit but does not fully meet PLOS ONE’s publication criteria as it currently stands. Therefore, we invite you to submit a revised version of the manuscript that addresses the points raised during the review process.

We would appreciate receiving your revised manuscript by Nov 07 2019 11:59PM. To enhance the reproducibility of your results, we recommend that if applicable you deposit your laboratory protocols in protocols.io, where a protocol can be assigned its own identifier (DOI) such that it can be cited independently in the future. For instructions see: http://journals.plos.org/plosone/s/submission-guidelines#loc-laboratory-protocols

We look forward to receiving your revised manuscript.

Kind regards,

Nülüfer Erbil, Ph.D, Prof.

Academic Editor

PLOS ONE

Journal Requirements:

2.  In your Methods section, please provide additional information about the participant recruitment method and the demographic details of your participants. Please ensure you have provided sufficient details to replicate the analyses such as: a) a description of any inclusion/exclusion criteria that were applied to participant recruitment; and b) a statement as to whether your sample can be considered representative of a larger population.

3. Please supply a copy of the questionnaire that was administered to study participants as a new supplementary information figure.

4. Please amend your current ethics statement to confirm that your named institutional review board or ethics committee specifically approved this study.

5. We suggest you thoroughly copyedit your manuscript for language usage, spelling, and grammar. If you do not know anyone who can help you do this, you may wish to consider employing a professional scientific editing service.  

Additional Editor Comments (if provided):

Reviewers' comments:

Reviewer's Responses to Questions

**Comments to the Author**

1. Is the manuscript technically sound, and do the data support the conclusions?

Reviewer #1: Yes

Reviewer #2: Partly

2. Has the statistical analysis been performed appropriately and rigorously? 

Reviewer #1: No

Reviewer #2: Yes

3. Have the authors made all data underlying the findings in their manuscript fully available?

Reviewer #1: Yes

Reviewer #2: No

4. Is the manuscript presented in an intelligible fashion and written in standard English?

Reviewer #1: No

Reviewer #2: No

5. Review Comments to the Author

Reviewer #1: Utilization of Cervical Cancer Screening and Associated Factors among Women in Debremarkos town, Amhara Region, Northwest Ethiopia: Community based cross sectional study

Dear Editor, thank you for the opportunity to review this manuscript. Well done to the authors for sharing their work.

The paper presents data from a study conducted on utilization of cervical cancer screening and associated factors among women in Debremarkos town, Amhara Region, Northwest Ethiopia through a community based cross sectional study. However the study is much broader in scope that the title suggests as it includes knowledge and attitudes which I think should come out in the title. The work also requires a thorough edit as there are quite a number of grammatical and punctuation errors. There is also need to improve paragraphing throughout the write up to better present the manuscript. Here below are my specific comments and suggestions.

Suggested title: knowledge, attitudes and utilisation of cervical cancer screening services in Debremarkos town, Amhara Region, Northwest Ethiopia: A cross sectional study

Abstract

Should be adjusted based on revisions in other sections including presenting results on knowledge and attitudes.

Please provide a list of key words that best describe your article

Introduction

Instead of disease say common malignancy.

About 266000 – include timeframe; sentence requires editing

In Ethiopia, 70;95 cases – is there a typo here?

Avoid one sentence paragraphs. In fact, many sentences are hanging. You need to combine related sentences to form clear paragraphs.

Visual Acetic Acid is VIA not VAI

Write study aim in past tense. Let it also encompass knowledge and attitudes.

Methods

Not clear why an unconventional margin of error of 3.5% was used and not the usual 5%. This is so much an indication of sample size having been calculated afterwards. Please explain.

Name the local language(s) that questionnaires were translated to especially that this is a multi-lingual region.

Name the dependent variable(s).

How were study respondents selected through the various levels? What type of sampling was used? What were the inclusion and exclusion criteria? And where were the interviews carried out from?

What manipulations were done to form composite variables for knowledge and attitudes?

How was utilisation defined and how was this too manipulated for further analysis.

Results

Only describe key results and refer readers to the table for details. There is no need to mention every result in the write up as currently is.

Make the occupation categories informative. What does self-employed mean? Is it that they operate small businesses? How about private employees, what are these? Let these be made clear by using easy to understand descriptors so that readers will be on the same page.

HH income – indicate currency.

Reproductive and behavioural characteristics

Started sexual intercourse for the first time >> first had sexual intercourse at 16.

Many of the variables used for this section are vague, not well described and don’t add any information to the study. For example what do you mean by history of MSPs? In what period? Is it even informative? History of smoking? STDs? Within what time period? Why is this relevant?

Knowledge

The knowledge section is prominent and thus should be included within the study objectives or aim.

The knowledge questions too are vague. First of all indicate in the methods how knowledge and attitudes were scored and what you mean by being knowledgeable or positive attitude? Also show these results within your tables.

Cervical cancer screening can be practiced – what does this mean? What information can you take away from this? This is vague.

Convert knowledge questions into phrases. E.g. instead of ‘do you know the symptoms of cervical cancer’ we can have ‘knew symptoms of cervical cancer’ though why would such a question be relevant when you have a question on some of the symptoms themselves?

Cervical cancer is a killer disease. Of course it is. Don’t you think this is vague when asked this way?

Is cervical cancer curable disease if detected early? This is a leading question especially that you add the early detection. Question should have been only about it being curable.

Attitudes

Attitude too comes out prominently and so should feature in the study objectives / aim and title.

First two questions are same / similar as this questionnaire is responded to by women.

What is a family disease? Please make clear exactly what you were seeking and communicate it clearly.

What do you mean by hazardous to your health?

What is wellbeing and how really was this translated to the respondents. Or was it health? But what was the relevance of this very general question?

What is favourable attitude? What constituted it? Make this clear and present within your results.

Utilisation of cervical cancer screening

Where is the figure of 5.4%? Include within your tables. What was the definition of utilisation? Provide this early in the methods section.

Do not describe the analysis again. Need to omit the whole first paragraph which is full of repetitions.

Describe results from both bivariate and multivariate analysis. The section can also be made more informative by indicating the direction of the association.

Many of the results from further analysis are not informative at all and seem forced because of the very small numbers involved. You have cells with a single value. This is not sufficient to support further analysis and is responsible for some of the wide confidence intervals. You need to categorise most of these variables and repeat the analysis to present informative results. There is no need to have several categories with such small numbers. Otherwise, some covariates with very small numbers could be omitted. Also, do not forget that many variables especially for knowledge and attitudes are vague and you should carefully think whether they should actually be included in your model. Also the combined knowledge and attitude variables should be described earlier.

The asterisks used as footnotes in the tables are not used within the table.

Discussion

Do not repeat results within the discussion. First paragraph does not even show the comparison. Paragraphing, grammar and sentence structured all need to be improved.

Very long sentences should be revised and rewritten.

Many of the provided explanations for results are vague and do not show good grounding in the available cervical cancer literature. There is no evidence that exposure to risk factors leads to utilisation of screening. It could be more due to increased exposure to information and maybe increasing risk with age and thus more interest or contact with health facilities.

Marital status association can’t be explained to be due to having MSPs. This could be linked to preceding paragraph of age as widowed or divorced women are more likely to be older. Also, young ones are more likely educated… etc.

The explanation of age at first sexual intercourse seems forced and is vague. You can’t claim that symptoms of the disease would appear earlier in such a group. Also, you do not show any evidence between risk and likelihood to screen.

Some of the concepts being discussed can be merged and well synthesized and presented in shorter clear paragraphs than simply listing the results and forcing their explanations.

Knowledge and attitudes were presented in the results. Similarly, they should be discussed. Conclusion should also capture these.

Recommendation

Manuscript requires substantial revisions before it is sent for another round of review. I hope the above comments will help improve it.

Reviewer #2: -Formatting of the manuscript should follow the journal guideline

-Table, figures, main body should be according to the guideline

-Declaration sections also need some modification

6. PLOS authors have the option to publish the peer review history of their article (what does this mean?). If published, this will include your full peer review and any attached files.

Reviewer #1: No

Reviewer #2: No

---

## [Author Response · Author response to Decision Letter 0]

14 Nov 2019

Dear reviewer 

Thank you for your constructive comments on our manuscript. Again your suggestion towards title modification is also right. 

But from the outset, our outcome variable is "utilization of cervical cancer screening ". Knowledge and attitude are our independent variables so that variables that influence knowledge and attitude not fully included in the questionnaire. Again, during analysis, we have not used Likert scale analysis for attitude since we have used it as an independent variable. Because of these reasons, we have decided to put the original title (Utilization of Cervical Cancer Screening and Associated Factors among Women in Debremarkos town, Amhara Region, Northwest Ethiopia: Community based cross-sectional study).

We have used a 3.5 % margin of error to increase sample size and the rule of thumb says if ‘P-value' less than 20% and greater than 80% we can use margin of error from 2% to 4%.

Since it has yes/no outcome, the utilization of cervical cancer screening is analyzed by using binary logistic regression. 

The knowledge question was changed to binary (1=yes and no=0). Then calculated the mean value and women who answered knowledge questions score of mean value or above were considered as knowledgeable. 

The attitude question was also changed to binary (1=agree and 0=disagree + indifferent). Then calculated the mean value and women who answered attitude questions a score of mean value or above were considered to have a favorable attitude.

Self employee means individuals who are doing their own small business and private employee means individuals who are salaried in the private sector. As you have recommended, the definitions are described in the bracket. 

History of MSPs, History of smoking, and SDs are adapted from different literatures. History of MSPs and History of smoking have been operated. They are relevant because different they are associated with cervical cancer screening in different studies.

In attitude, the first two questions seem similar in concept but, women may not think as they are susceptible to cervical cancer even though they think as others are susceptible. That is why the two questions have different figure (25.2% versus 24.6% )

We have used “Cervical cancer screening can be practiced” to say ‘All eligible mothers can have cervical cancer screening without complication’ 

“Cervical cancer is a killer disease". We know as it is a killer disease but since it is a community-based study, women may not know whether it is a killer disease or not. So, it may be appropriate. 

A family disease means a disease that can be transmitted genetically. 

Hazardous to your health means cervical cancer may be dangerous which complicates women’s life. 

Wellbeing is used to show the benefit of cervical cancer screening to health. Even though it is a general question to assess the attitude of women towards cervical cancer screening. Women may think screening by itself induces cervical cancer. 

We have cells with a single value. The main reason is there is a small number of data that were found based on ethnicity (after merging of Oromo and Gurage, only one person was utilized screening) so, we could not merge it further.

---

## [Decision Letter · Decision Letter 1]

9 Dec 2019

PONE-D-19-16524R1

Utilization of Cervical Cancer Screening and Associated Factors among Women in Debremarkos town, Amhara Region, Northwest Ethiopia: Community based cross sectional study

PLOS ONE

Dear mr yeserah,

Thank you for submitting your manuscript to PLOS ONE. After careful consideration, we feel that it has merit but does not fully meet PLOS ONE’s publication criteria as it currently stands. Therefore, we invite you to submit a revised version of the manuscript that addresses the points raised during the review process.

We would appreciate receiving your revised manuscript by Jan 23 2020 11:59PM. To enhance the reproducibility of your results, we recommend that if applicable you deposit your laboratory protocols in protocols.io, where a protocol can be assigned its own identifier (DOI) such that it can be cited independently in the future. For instructions see: http://journals.plos.org/plosone/s/submission-guidelines#loc-laboratory-protocols

We look forward to receiving your revised manuscript.

Kind regards,

Nülüfer Erbil, Ph.D, Prof.

Academic Editor

PLOS ONE

Reviewers' comments:

Reviewer's Responses to Questions

**Comments to the Author**

1. If the authors have adequately addressed your comments raised in a previous round of review and you feel that this manuscript is now acceptable for publication, you may indicate that here to bypass the “Comments to the Author” section, enter your conflict of interest statement in the “Confidential to Editor” section, and submit your "Accept" recommendation.

Reviewer #1: (No Response)

Reviewer #2: All comments have been addressed

2. Is the manuscript technically sound, and do the data support the conclusions?

Reviewer #1: Partly

Reviewer #2: Partly

3. Has the statistical analysis been performed appropriately and rigorously? 

Reviewer #1: No

Reviewer #2: Yes

4. Have the authors made all data underlying the findings in their manuscript fully available?

Reviewer #1: No

Reviewer #2: Yes

5. Is the manuscript presented in an intelligible fashion and written in standard English?

Reviewer #1: No

Reviewer #2: Yes

6. Review Comments to the Author

Reviewer #1: Dear Editor,

The manuscript has improved but I believe these few issues should also be considered:

• Throughout the manuscript, use women instead of mothers. e.g in sampling technique, table 3 etc.

• Have another look at paragraph 3 and 4 in the discussion. The writing does not bring out any key message and is unnecessarily long. In fact, have another read of the manuscript and address the typos and omissions, grammar and punctuation especially in the discussion.

• You may also refer to this recent publication very similar to yours in the same area to further support your discussion.

Uptake of pre-cervical cancer screening and associated factors among reproductive age women in Debre Markos town, Northwest Ethiopia, 2017 https://bmcpublichealth.biomedcentral.com/articles/10.1186/s12889-019-7398-5

Reviewer #2: It is better to consider the comments given in the main document. each of the comment given is very critical and should be addressed

7. PLOS authors have the option to publish the peer review history of their article (what does this mean?). If published, this will include your full peer review and any attached files.

Reviewer #1: No

Reviewer #2: No

---

## [Author Response · Author response to Decision Letter 1]

12 Dec 2019

thank you for your constructive comments.

---

## [Decision Letter · Decision Letter 2]

14 Jan 2020

PONE-D-19-16524R2

Utilization of Cervical Cancer Screening and Associated Factors among Women in Debremarkos town, Amhara Region, Northwest Ethiopia: Community based cross sectional study

PLOS ONE

Dear mr yeserah,

Thank you for submitting your manuscript to PLOS ONE. After careful consideration, we feel that it has merit but does not fully meet PLOS ONE’s publication criteria as it currently stands. Therefore, we invite you to submit a revised version of the manuscript that addresses the points raised during the review process.

We would appreciate receiving your revised manuscript by Feb 28 2020 11:59PM. To enhance the reproducibility of your results, we recommend that if applicable you deposit your laboratory protocols in protocols.io, where a protocol can be assigned its own identifier (DOI) such that it can be cited independently in the future. For instructions see: http://journals.plos.org/plosone/s/submission-guidelines#loc-laboratory-protocols

We look forward to receiving your revised manuscript.

Kind regards,

Nülüfer Erbil, Ph.D, Prof.

Academic Editor

PLOS ONE

Reviewers' comments:

Reviewer's Responses to Questions

**Comments to the Author**

1. If the authors have adequately addressed your comments raised in a previous round of review and you feel that this manuscript is now acceptable for publication, you may indicate that here to bypass the “Comments to the Author” section, enter your conflict of interest statement in the “Confidential to Editor” section, and submit your "Accept" recommendation.

Reviewer #1: All comments have been addressed

Reviewer #2: All comments have been addressed

2. Is the manuscript technically sound, and do the data support the conclusions?

Reviewer #1: Partly

Reviewer #2: Partly

3. Has the statistical analysis been performed appropriately and rigorously? 

Reviewer #1: No

Reviewer #2: Yes

4. Have the authors made all data underlying the findings in their manuscript fully available?

Reviewer #1: Yes

Reviewer #2: Yes

5. Is the manuscript presented in an intelligible fashion and written in standard English?

Reviewer #1: Yes

Reviewer #2: No

6. Review Comments to the Author

Reviewer #1: (No Response)

Reviewer #2: Please make sure that the review questions need to be considered during the re sending the document!

7. PLOS authors have the option to publish the peer review history of their article (what does this mean?). If published, this will include your full peer review and any attached files.

Reviewer #1: No

Reviewer #2: No

---

## [Author Response · Author response to Decision Letter 2]

8 Feb 2020

dear reviewers and and editors, thank you for your comments. we are happy if you help us for this manuscript publication. since this research finding is presented with real data, we are happy if it is published as soon as possible.

 thanks

---

## [Decision Letter · Decision Letter 3]

23 Mar 2020

Utilization of Cervical Cancer Screening and Associated Factors among Women in Debremarkos town, Amhara Region, Northwest Ethiopia: Community based cross sectional study

PONE-D-19-16524R3

Dear Dr. Aynalem,

We are pleased to inform you that your manuscript has been judged scientifically suitable for publication and will be formally accepted for publication once it complies with all outstanding technical requirements.

With kind regards,

Nülüfer Erbil, Ph.D, Prof.

Academic Editor

PLOS ONE

Additional Editor Comments (optional):

Reviewers' comments:

Reviewer's Responses to Questions

**Comments to the Author**

1. If the authors have adequately addressed your comments raised in a previous round of review and you feel that this manuscript is now acceptable for publication, you may indicate that here to bypass the “Comments to the Author” section, enter your conflict of interest statement in the “Confidential to Editor” section, and submit your "Accept" recommendation.

Reviewer #1: All comments have been addressed

Reviewer #2: All comments have been addressed

2. Is the manuscript technically sound, and do the data support the conclusions?

Reviewer #1: Partly

Reviewer #2: Yes

3. Has the statistical analysis been performed appropriately and rigorously? 

Reviewer #1: No

Reviewer #2: Yes

4. Have the authors made all data underlying the findings in their manuscript fully available?

Reviewer #1: Yes

Reviewer #2: Yes

5. Is the manuscript presented in an intelligible fashion and written in standard English?

Reviewer #1: No

Reviewer #2: Yes

6. Review Comments to the Author

Reviewer #1: (No Response)

Reviewer #2: The author addressed all the comments given previously and incorporated with the new revised version. But my concern is in discussion section many of the literatures used had a different approaches of study and then how the different modalities of research finding will be equivalently compared with current study? if the author make this confusion, I am agree with the stand of this manuscript to be published.

7. PLOS authors have the option to publish the peer review history of their article (what does this mean?). If published, this will include your full peer review and any attached files.

Reviewer #1: No

Reviewer #2: No

---

## [Editor Report · Acceptance letter]

25 Mar 2020

PONE-D-19-16524R3 

Utilization of cervical cancer screening and associated factors among women in Debremarkos town, Amhara Region, Northwest Ethiopia: Community based cross-sectional study 

Dear Dr. Aynalem:

I am pleased to inform you that your manuscript has been deemed suitable for publication in PLOS ONE. Congratulations! Your manuscript is now with our production department. 

With kind regards,

on behalf of

Mrs. Nülüfer Erbil 

Academic Editor

PLOS ONE